

# Field-scale soil moisture bridges the spatial-scale gap between drought monitoring and agricultural yields

Noemi Vergopolan[1], Sitian Xiong[2], Lyndon Estes[2], Niko Wanders[3], Nathaniel W. Chaney[4], Eric F. Wood[1], Megan Konar[5], Kelly Caylor[6,7], Hylke E. Beck[1], Nicolas Gatti[8], Tom Evans[9], and Justin Sheffield[10]

[1]Civil and Environmental Engineering Department, Princeton University, USA
[2]School of Geography, Clark University, USA
[3]Department of Physical Geography, Faculty of Geosciences, Utrecht University, The Netherlands
[4]Department of Civil and Environmental Engineering, Duke University, USA
[5]Civil and Environmental Engineering Department, University of Illinois at Urbana-Champaign, USA
[6]Department of Geography, University of California, Santa Barbara, USA
[7]Bren School of Environmental Science and Management, University of California, Santa Barbara, USA
[8]Department of Agricultural and Consumer Economics, University of Illinois at Urbana-Champaign, USA
[9]School of Geography, Development and Environment, University of Arizona, USA
[10]School of Geography and Environmental Science, University of Southampton, Southampton, UK

**Correspondence:** Noemi Vergopolan (noemi.v.rocha@gmail.com)

**Abstract.** Soil moisture is highly variable in space, and its deficits (i.e., droughts) plays an important role in modulating crop yields and its variability across landscapes. Limited hydroclimate and yield data, however, hampers drought impact monitoring and assessment at the farmer field-scale. This study demonstrates the potential of field-scale soil moisture simulations to advance high-resolution agricultural yield prediction and drought monitoring at the smallholder farm field-scale. We present a multi-scale

modeling approach that combines HydroBlocks, a physically-based hyper-resolution Land Surface Model (LSM), and machine learning. We used HydroBlocks to simulate root zone soil moisture and soil temperature in Zambia at 3-hourly 30-m resolution. These simulations along with remotely sensed vegetation indices, meteorological conditions, and data describing the physical properties of the landscape (topography, land cover, soil properties) were combined with district-level maize data to train a random forest model (RF) to predict maize yields at the district- and field-scale (250-m) levels. Our model predicted yields with

a coefficient of variation ($R^2$) of 0.61, Mean Absolute Error (MAE) of 349 $\mathrm{kg\,ha^{-1}}$, and mean normalized error of 22%. We captured maize losses due to the 2015/2016 El Niño drought at similar levels to losses reported by the Food and Agriculture Organization (FAO). Our results revealed that soil moisture is the strongest and most reliable predictor of maize yield, driving its spatial and temporal variability. Consequently, soil moisture was also the most effective indicator of drought impacts in crops when compared with precipitation, soil and air temperatures, and remotely-sensed NDVI-based drought indices. By combining

field-scale root zone soil moisture estimates with observed maize yield data, this research demonstrates how field-scale modeling can help bridge the spatial scale discontinuity gap between drought monitoring and agricultural impacts.





# 1 Introduction

Droughts can significantly impact crop production, with implications for food security, particularly in smallholder farming systems (Kristjanson et al., 2012; Guilpart et al., 2017). The impacts of droughts to agricultural production remain difficult to

quantify, especially in developing regions, where data is generally scarce and crop production can be highly variable due to high climate variability, heterogeneous landscapes, and variable farming capacities (Lobell, 2013; Donaldson and Storeygard, 2016). Challenges in understanding the precise impact of drought on crop yields exist because of the lack of high-quality data and appropriate drought metrics (Sutanto et al., 2019; Beza et al., 2017; Sadri et al., 2020). These data limitations may lead to predictions of yields, drought impacts, and, consequently, agricultural management insights that do not accurately capture the

impacts at the farm-level. Improving our understanding of how drought impacts agriculture across spatiotemporal scales would improve the robustness of agricultural drought risk frameworks, and leverage the government's ability to design and implement policies to reduce crop losses.

For example, during the 2015/2016 El Niño, one of the strongest on record (Kintisch, 2016), drought severely impacted Sub-Saharan Africa (SSA). Crop yields dropped 20% in Zambia (Alfani et al., 2019), 63% in Somalia, 50% in Ethiopia,

49% in Zimbabwe, 31% in Swaziland (FAO, 2016a), and 40% in Malawi (FAO, 2016b), leading the region to a state of emergency due to food shortages. Despite the evident severity, in Malawi, for example, satellite-based rainfall drought indices identified that only 21,000 farmers were affected by the drought, while, in reality, survey-based assessments identified that 6.5m farmers were impacted (Economist, 2016). Although rainfall has a historical significant contribution to monitoring droughts and agricultural impacts (Zargar et al., 2011; Hao and Singh, 2015; Van Loon et al., 2016), inconsistencies, as such, emerged because

rainfall-based indices do not account for the extreme heat associated with drought. By not accounting for the plant-soil-water dynamics and interactions with the landscape (Peichl et al., 2018; Franz et al., 2020), rainfall-based metrics often do not directly reflect how much water is available to plants. In fact, during the 2015/2016 El Niño, the extreme heat led to insufficient soil moisture in the rooting zone for the plants to meet the higher than normal atmospheric moisture demand, which increased the drought impacts above and beyond the deficit in rainfall supply (Kintisch, 2016; Wanders et al., 2017).

For these reasons, hydrological variables such as soil moisture and evapotranspiration are a more direct proxy of the water available in the root zone to plants. In fact, soil moisture has been shown to better predict agricultural drought impacts than precipitation and air temperature measures (Xia et al., 2014; Bachmair et al., 2016). However, in-situ soil moisture measurements or information on the root zone are virtually non-existent in most of the developing world (Karthikeyan et al., 2017). Satellites can provide global information on soil moisture with a 2-3 day revisit time, but they have limited spatial resolution (e.g., 9 km

for the NASA SMAP-Enhanced product) and can only measure the upper 5 cm of the soil. Thus, remote sensing based estimates of soil moisture fall short in representing conditions at the scale of agricultural fields ($\sim$ 1-10 ha) and crop rooting zones (10-150 cm). Consequently, few studies have been able to quantify the relationship between field-scale soil moisture deficit (i.e., drought) and crop yield (for a review, see Karthikeyan et al., 2020).

To aid drought assessments, data on crop yields are usually estimated through self-reported field surveys. However, these

are time-consuming, expensive, and often suffer from sampling and reporting errors (Paliwal and Jain, 2020; Gourlay et al.,





2019). To compensate for these errors, survey data are generally aggregated to the scale of administrative units, which masks the heterogeneity of yields that exists across small-scale farms (0-5 ha; Jayne et al., 2016). Previous studies indicate that there is a large variability between and within fields, which is substantially masked by aggregation (e.g., Lobell et al., 2007; Franz et al., 2020). This variability is due to spatiotemporal variations in weather (which occurs at kilometer scales or finer), diversity in

farm management strategies, and the spatial variability in the landscape (including topography and soils; that can act at the meter scale). These spatiotemporal variations propagate into small scale variations in hydrological variables and fluxes, such as soil moisture and evapotranspiration (Crow et al., 2012; Chaney et al., 2018). Variations in planting date, cultivar choice, and fertilizer/pesticide applications also create inter-field yield heterogeneity for fields with similar environmental attributes. It is, therefore, difficult to interpret spatially aggregated yields because they average out important aspects of the spatial variability in

the underlying data. Unknowing the field-scale yield and drought impacts variability also complicates how drought policies are designed and implemented, especially because individual fields and farmers may respond differently and require different interventions during a drought. Thus, characterizing the spatiotemporal dynamics of agricultural yields and droughts at the farm scale (1–250 m resolution) is critical to better understand the field-scale circumstances and to better guide on-the-ground interventions.

There is a long and diverse legacy of attempts to develop models that can predict how agricultural yields respond to weather and climate. These include both process-based (e.g., Jones et al., 2003; Keating et al., 2003) and empirical approaches based on statistical (Lobell and Burke, 2010) and machine learning methods Chlingaryan et al. (2018). These approaches are mostly based on predictors related to precipitation, temperature, and satellite-derived vegetation indices (VIs), which can help resolve the spatiotemporal variability in yields but are only partially correlated with actual yields (e.g., Lobell et al., 2007; Enenkel

et al., 2018). Ideally, vegetation greenness can capture the combined influence of hydroclimatic variability (Koster et al., 2014; Adegoke and Carleton, 2002) and agricultural management activities (e.g., irrigation and fertilization Deines et al., 2017; Estel et al., 2016; Chen et al., 2018). However, VIs are derived from visible-infrared satellite sensors that are impacted by a number of factors that can undermine yield estimates, such as long revisit times (1–2 weeks), cloud contamination, and saturation at high values (e.g. NDVI, Azzari et al., 2017; Gu et al., 2013), which limits its application.

In this study, we present a multi-scale framework that combines hyper-resolution land surface modeling and machine learning to obtain field-scale maize yield estimates and gain insight into the relationships between drought indices and yield variability. Specifically, we used the HydroBlocks land surface model (Chaney et al., 2016; Vergopolan et al., 2020) to simulate root zone soil moisture and surface temperature at a high spatial and temporal resolution (30-m, 3-h interval) over a long duration (1981–2018). We combine these field-scale measures with meteorological variables, remotely sensed vegetation indices, and

several other socioeconomic and physical measures with a random forest model (RF; Breiman, 2001) to predict annual maize yields at both district and field scales for Zambia (750,000 $\mathrm{km}^2$), a Southern African country that is exposed to substantial climate variability and where much of the population still depends on small-scale agriculture (Zhao et al., 2018). We use this modeling framework to answer the following questions:

(i) What are the most influential drivers of maize yield variability, and how do hydrological- versus meteorological-based

predictors contribute to yield predictions?





(ii) What is the field-scale variability of the predicted yields?

(iii) How and what drought conditions lead to yields losses at the field-scale?

This study shows the critical role of soil moisture in modulating maize yields, outperforming precipitation, temperature, and vegetation index predictors. We demonstrate how droughts can impact yields differently across the landscape, and how field-scale soil moisture percentiles can effectively capture drought-associated crop losses.

## 2 Data and Methods

### 2.1 Study area

Our study focuses on Zambia, which is broadly representative of the smallholder-dominated farming systems in many of Africa's savanna regions, which are also beginning to undergo a period of rapid development in which agriculture will play a key part (Searchinger et al., 2015). Savannas drylands are characterized by strong rainfall seasonality and often high inter- and intra-seasonal rainfall variability, which has important consequences for food security (Lehmann and Parr, 2016; Scanlon et al., 2005; D'Odorico and Bhattachan, 2012). Zambia's annual precipitation ranges from 1400 mm in the north to 700 mm in the south, with an annual mean air temperature of 20 °C that rises to 25 to 30 °C during the growing season. The wet season is generally from October to April and the dry season is from June to September, with the date of rainy season onset earlier in the northern part of the country than in the south. The sowing period extends approximately from October to December, the growing period extends from November to May, and the harvesting season extends from April to June (Waldman et al., 2019). In 2013, Zambia consisted of 72 districts (118 districts in 2020 after subdivision of some districts) with an average area of 10,450 $\mathrm{km}^2$ and an average agricultural area of 3,310 $\mathrm{km}^2$ per district. Figure S1 in the Supplemental Information (SI) shows the districts and land cover. While 35.8% of the Zambian agricultural area is of small-size farms (0–5 ha) and 53.0% is of medium-size farms (5–100 ha), 78.8% of the land is owned by smallholder farmers with farms of size 0–5 ha (Jayne et al., 2016). Farming is the primary livelihood activity for 85% of the population, as is the case with many other SSA countries (GYGA, 2020). Irrigation systems are mostly absent in the small-scale farming sector, with agriculture heavily relying on rainfall (Mason and Myers, 2013). Maize is Zambia's key commodity with a potential yield of 12000 $\mathrm{kg\,ha}^{-1}$, the same as in the United States, but with actual yields of, in average, 1600 $\mathrm{kg\,ha}^{-1}$ (GYGA, 2020).

### 2.2 Hyper-resolution land surface modeling

Hydroblocks is a field-scale Land Surface Model (LSM) that considers high-resolution ancillary datasets (soil properties, topography, and land cover at 30–250 m resolution) as drivers of landscape spatial heterogeneity (Chaney et al., 2016). HydroBlocks leverages the repeating patterns that exist over the landscape (i.e., the spatial organization) by clustering areas of assumed similar hydrologic behavior into Hydrological Response Units (HRUs). The identification of these HRUs and their spatial interactions allows the modeling of hydrological, geophysical, and biophysical processes at the field-scale over regional to continental extents. The core of HydroBlocks is the Noah-MP LSM (Niu et al., 2011) single-column land surface scheme.





HydroBlocks applies Noah-MP in an HRU framework to explicitly represent the spatial heterogeneity of surface processes down to field scale. At each time step, the land surface scheme updates the hydrological states at each HRU; and the HRUs dynamically interact laterally via subsurface flow. HydroBlocks implements a multi-scale hierarchical scheme that operates at
several spatial scales identified for the underlying hydrological, geophysical, and biophysical processes (Chaney et al., 2018):

(a) Catchments: defined by topography and serve as the boundary for surface flows;

(b) Characteristic hillslopes: defined by topography and environmental similarity;

(c) Height bands: defined by the height above nearest drainage and define the primary flow directions and surface temperature gradient;

(d) HRUs: defined by multiple soil/vegetation/land cover characteristics and serve as the smallest modeling units.

With this hierarchical setup, HydroBlocks handles mass/energy exchanges within a modeling unit (at a certain scale) separately from the exchanges between units at that scale. This enables full and realistic horizontal coupling while ensuring computational efficiency.

We deployed the HydroBlocks to simulate root zone soil moisture and soil temperature from surface to 1.5 m depth at 3-h
30-m resolution between 1981-2018. As data inputs, we used hourly 9-km meteorological inputs from ERA5-Land (rainfall, 2-m air temperature, longwave radiation, shortwave radiation, wind, surface pressure, and specific humidity derived from dew point). ERA5-Land is a state-of-the-art global reanalysis product that was chosen because of its high spatial resolution and overall good performance in representing rainfall and soil moisture dynamics (Beck et al., 2020). To parameterize the Hydroblocks, we also used 30-m topography (SRTM; Farr et al., 2007); 20-m land cover type (ESA-CCI); and 250-m soil properties (SoilGrids; Hengl
et al., 2017) to derive the soil hydraulic parameters via pedotransfer functions. Previous validation work compared HydroBlocks soil moisture simulations with in-situ observations over similar climates, and results showed that the model represents the spatial and temporal variability of in-situ soil moisture measurement as field-scale and regional scale, with comparable performance to satellite estimates (Vergopolan et al., 2020; Cai et al., 2017), providing reasonable confidence in the variability of the yield estimates derived in this study.

**2.3  Modeling maize yields at district-scale and mapping yields at field-scale**

To model and predict maize yield dynamics at the field-scale, we trained a Random Forests (RF) model on district-level survey-based maize yield data, high-resolution HydroBlocks simulations, remotely sensed vegetation index, meteorological reanalysis data, and other static information on the landscape. Section 2.3.1 presents the datasets and predictors descriptions. Section 2.3.2 presents the RF model setup and evaluation. Section 2.3.3 presents the recursive feature elimination approach
applied to select and rank predictors. Lastly, Section 2.3.4 presents how the RF model was deployed to predict annual maize yields in Zambia at the field-level, and the analysis performed to assess the yields spatial variability.





**Table 1.** List of the datasets and the predictors derived to train the random forest model

| Dataset | References | Predictors description | Type |
|---|---|---|---|
| Land Cover | 20-m ESA-CCI (2016) | The land cover classification was used to identify cropland areas (assuming that all cropland produce maize) and the percentage coverage of non-agricultural cover types in 250-m pixels. | Static |
| Topography | 30-m SRTM (Farr et al., 2007) | From elevation we derived slope, aspect, topographic index, and the height above the nearest drainage. | Static |
| Soil properties | 250-m SoilGrids (Hengl et al., 2017) | Clay, sand, silt, and organic matter content, and the following variables derived using pedo-transfer functions (Saxton and Rawls, 2006): soil saturation, field capacity, wilting point, soil water storage, and hydraulic conductivity calculated using pedotransfer functions. | Static |
| Climatology:<br>– Precipitation<br>– Air temperature | 1-km WorldClim2 (Fick and Hijmans, 2017) | Climatology of air temperature and precipitation during the growing season. | Static |
| Socioeconomic:<br>– GDP<br>– Population | 1-km 2010 GDP (Bank, 2012) & 2010 population (CIESIN, 2017) | GDP and population density as proxies for access to finance, technology, and infrastructure. | Static |
| Vegetation index:<br>– NDVI | 250-m MODIS MOD13Q1 (2000–2018) | Maximum growing season NDVI, date of maximum NDVI, and seasonal NDVI integrals calculated by using a smoothing function to fill the missing NDVI values and to remove outliers followed by a Savitzky-Golay (SG) filter (Chen et al., 2004). | Dynamic |
| Meteorological:<br>– Precipitation<br>– Air temperature | 9-km monthly ERA5-Land (C3S, 2019) | Growing season and the monthly average, minimum, and maximum estimates of air temperature and precipitation. | Dynamic |
| Hydrological:<br>– Soil moisture<br>– Soil temperature | 30-m monthly HydroBlocks (this study) | Seasonal and monthly average, minimum, and maximum root zone soil moisture, relative root zone soil moisture, and soil temperature from the HydroBlocks LSM. | Dynamic |

### 2.3.1 Training data and feature engineering

To train the RF model, we used the Post-Harvest Surveys (PHS) database of maize yields ($km^2$). This database comprises household survey data of ~13,000 farmers' self-reported harvested maize (kg or total bags of crop) and respective cultivated area (ha). The data was collected at the end of each harvesting season by Zambia's Central Statistical Office (a division of the Ministry of Agriculture and Livestock) from 1991 to 2005, 2007, 2008, 2011 to 2014. Due to privacy and data uncertainties, the observations were only available aggregated to the district-level (~10,000 $km^2$). In this work, to match the period of availability for remotely sensed predictors, we used the PHS data from 2000–2018. This resulted in a total of 527 observations from 70 districts and 8 years. The data were randomly split, with 80% used for training and 20% used for model evaluation. To train the model, we used a range of static and dynamic variables with time, as described in Table 1.

From these static and dynamic predictors, we identified 103 initial predictors. One of the challenges of working with multi-source and multi-scale datasets is the spatial scale mismatch of input data (20-m to 9-km resolution). To calculate the value of each predictor, we masked out the non-cropland pixels and calculated the area-weighted average based on the cropland location/area in each district. Despite the large district areas, the use of these area-weighted averages at only the agricultural areas helped to remove the influence from the surrounding non-cropland areas (i.e., grasslands, water bodies, urban areas), while accounting for the spatial variability of each predictor in the district. Lastly, each predictor was normalized based on the maximum and minimum values.





### 2.3.2 Random forest regressor for yield modeling

Machine learning models have been widely applied for crop yield prediction (Chlingaryan et al., 2018). Random Forest (RF)
regressors are used with geospatial hydroclimate and satellite data to predict maize yields at fine-scales (Aghighi et al., 2018;
Khanal et al., 2018; Jeong et al., 2016; Folberth et al., 2019). Advantages of RF regression models are their high predictive
accuracy even when trained on small, nonlinear, and collinear datasets, their robustness to outliers, and their ability to avoid
overfitting (Breiman, 2001; Archer and Kimes, 2008; Wylie et al., 2019). Dealing with data collinearity is particularly important
for yield prediction because often the meteorological-, hydrological-, and vegetation-based predictors are interconnected (Archer
and Kimes, 2008; Wylie et al., 2019).

To identify the optimal RF architecture, we performed a grid-search on the possible combinations of relevant hyper-parameters:
number of trees, maximum depth of the trees, the minimum number of samples required to split an internal node, the minimum
number of samples required to be at a leaf node, and the number of bootstraps of predictors. The best hyper-parameter
combination was selected based on the average Mean Square Error (MSE) of a 3 fold cross-validation on the training sample.
We further evaluate the overall model's performance by calculating the Mean Absolute Error (MAE) and the Coefficient of
Determination ($R^2$) performance with the testing sample (20% of the data).

### 2.3.3 Predictor importance and selection

Using the district-level PHS yield data and the RF model, we performed a Recursive Feature Elimination (RFE) analysis to (i)
identify and rank the most important predictors of maize yields, and (ii) to determine which of the predictors could be removed.
Removing non-predictive variables is particularly helpful as it can improve model accuracy, it mitigates the model's tendency to
overfit, while the smaller data volume reduces the computational cost.

The RFE is an iterative process. At each iteration, the model is trained, the importance of each predictor calculated and ranked,
and the least important predictor is removed (Gregorutti et al., 2016). This process continues until a convergence criterion is
met. In our implementation, we used the $R^2$ between the RF-predicted yields and the testing set as the evaluation metric. The
importance of each predictor was calculated based on the difference between the $R^2$ of the model with the predictor and the
model without the predictor. This iterative process of retraining and assessing the relative importance of each predictor ensures
that the least important predictor is consistently removed and that discarded variables either do not contribute to or degrade
model performance. Variable removal ceased once this $R^2$ difference fell below 0.001 (see Table S1 for a full description of the
RFE process). The RFE process applied to the 103 predictors in the training set (section 2.3.1) resulted in the retention of 36
important predictors.

In addition to variable selection, we performed a sensitivity analysis to quantify the relative value of hydrology-based
predictors (soil moisture and soil temperature) and meteorology-based predictors (precipitation and air temperature) in predicting
district-level maize yields. For this experiment, we trained the RF model, applied RFE, and identified the most important
predictors of three different predictor sets, as shown in Table 2. For each case, we quantified the final $R^2$ performance of the
testing sample and the delta $R^2$ importance of each predictor. We then compared the change in performance with and without





meteorology- and hydrology-based predictors with respect to the benchmark. We also compared the relative importance of different predictors in each of the cases.

**Table 2.** Predictor sets applied to the maize yield modeling sensitivity experiments

| Case | Name | Description | Total predictors |
|---|---|---|---|
| 1 | All predictors | All the predictors as a benchmark | 103 |
| 2 | No meteorology | All the predictors except the precipitation and air temperature predictors | 70 |
| 3 | No hydrology | All the predictors except the soil moisture and soil temperature predictors | 61 |

### 2.3.4 Predicting maize yield at the field-scale

Previous work showed that RF models are able to successfully model fine-scale crop yields from coarse-scale physically-based
crop models estimates, assuming that the fine-scale predictors are representative of the fine-scale yield variability (Folberth et al., 2019). Similarly, we deployed the trained RF model to predict maize yields at 250-m resolution, considering that predictors and model parameters that represent the yield spatiotemporal dynamics at the district-scale can represent yield dynamics at the field-scale.

To this aim, we first identified all 250-m grid cells that have at least 50% cropland coverage according to our fractional land
cover map derived from the ESA-CCI data (Table 1, Figure S1), which resulted in ∼1M grid cells. Since maize is Zambia's dominant crop (Ng'ombe, 2017), we assume that all the identified cropland areas are maize fields. Similarly to the training and testing set, we used the location/area of the cropland fields to calculate the area-weighted average of each predictor. We used our best RF model to predict the annual maize yield at each 250-m grid cell over Zambia between 2000 to 2018.

To characterize the spatiotemporal distribution of field-scale maize yields estimates derived from the RF model we generated
maps of mean annual yields and their temporal coefficient of variation. We also calculated maps of maize yield trends between 2000–2018 to identify the locations where increases in yields were larger. As an example, to quantify the spatial distribution of losses in yield due to drought, we calculated the relative change in yields for the 2015/2016 El Niño season and compared it to FAO survey estimates (Alfani et al., 2019). Despite the sampling difference between the surveyed and predicted field-scale yields, we evaluate the estimated field-scale yields by comparing the field-scale yields aggregated to the district level with
district-level PHS survey yield data. We computed the temporal $R^2$ and MAE, and the mean spatial Pearson correlation.

*Spatial variability of field-scale maize yields and main predictors*

The spatial patterns and spatial variability in maize yields can be driven by hydroclimatic conditions, soil properties, topography, and also farmer management (such planting date, seed choice, use of fertilizers, irrigation, etc.). Consequently, droughts can impact the landscape differently. To quantify the strength of each predictor in driving the spatial variability in maize yields at the
local scale, we calculated the spatial Pearson correlation between the field-scale yields and the most important predictors. The





**Table 3.** List of the six drought impact indicators used in the crop losses analysis. Three are the most predictive variables (identified by the predictor importance analysis), and three are respectively commonly used indicators.

| Indicator conditions | Variables | |
|---|---|---|
| | Most predictive | Commonly used |
| Dry/wet | Root zone soil moisture (Apr.) | Precipitation (Apr.) |
| Cool/hot | Max. soil temperature (Oct.) | Max. air temperature (Oct.) |
| Plant health | Date of NDVI peak (season) | NDVI Integral (season) |

time series of spatial correlation was calculated for each year over the entire country, as well as for three smaller domains (of 50-km by 50-km) in the north, central, and south of the country.

### 2.4 Characterizing the relationship between field-scale yields and drought indicators

The predictor importance analysis (section 2.3.3) identified the most influential variables impacting maize yields at the district-
scale. To gain further insight into the potential effectiveness of these variables as drought impact indicators, we compared how they varied with the spatially co-located de-trended maize yields, hereafter called anomalies. This allows us to characterize the relationship between these indicators (e.g., dry and hot) versus local losses or gains in maize yields.

The drought indicator variables, shown in Table 3, were identified based on the three most important predictors (result from section 3.2) and three (respective) commonly used indicators. Temperatures and date of NDVI peak were used in their original
units. For NDVI integrals, for each grid, we used the temporal anomaly relative to the 2000-2018 mean. For soil moisture and precipitation, drought indices were constructed from monthly values and converted to the empirical percentile. The empirical percentile was calculated for each time step and grid cell and based on the monthly historical records (1981–2018) for that location/month (Sheffield, 2004). By using percentiles, rather than absolute values, drought events can be compared in space and time, despite their local characteristics.

By comparing how each drought indicator varied with the yield anomalies, this approach allowed us to characterize the relationship between these indicators versus local losses (or gains) in maize yields. We delineated potential drought thresholds for soil moisture and precipitation percentiles and quantified the mean expected yield losses associated with each threshold. Second, we compared how each of the drought indicators co-varied with each other and with maize yield anomalies. This allowed us to quantify what are the expected yield losses (or gains) under dry and hot versus wet and hot conditions, and to
identify in which conditions yield losses are driven by water deficits and/or temperature stress.

## 3   Results

### 3.1   Hydrological simulations at field-scale

HydroBlocks-simulated root zone soil moisture and soil temperature at 30-m resolution reveal substantial spatial variability at both the national and local scales. Figure 1 shows the mean April root zone soil moisture and October soil temperature





since these variables are the most important predictors of yield (section 3.2). At the national scale, wet and cooler conditions are observed in the northern parts of the country and near river valleys, while the south and southeast show distinctly dry and hot conditions. National and local scale variations (Figure 1 inset) reflect the interactions of meteorological conditions with the landscape, highlighting the influence of topography, soil properties, and vegetation cover on the spatial variability of hydrological processes.

## 3.2    District-level maize yield modeling and importance of predictors


The best-performing RF model was able to predict the values of the district-level yield data in the independent testing data set with a Mean Absolute Error (MAE) of 349 $\mathrm{kg\,ha^{-1}}$ and an $R^2$ of 0.61, with an MAE of 253 $\mathrm{kg\,ha^{-1}}$ and $R^2$ of 0.79 (Figure 2). The model captures the overall patterns of district-level maize yields, but with a tendency to underestimate maize yields above 2000 $\mathrm{kg\,ha^{-1}}$.

The recursive feature elimination process combined with the sensitivity analysis (Table 2) revealed that soil moisture and soil temperature variables were consistently the strongest predictors of yield (Figure 3). In the first two sensitivity tests, in which Hydroblocks variables were included, soil moisture and temperature variables provided 5 or 6 of the top 7 predictors, while removing soil moisture and temperature variables as predictors (case 3) resulted in the largest drop in overall model $R^2$ (0.09 and 0.11). On the other hand, removing meteorological variables resulted in relatively little loss of model explanatory power,

with an $R^2$ drop of only 0.02 between cases 1 and 2.

   In terms of specific variables, cases 1 and 2 both showed that the mean relative soil moisture in April was the strongest predictor of yield, followed by October and February soil temperatures, and then March soil moisture (Figure 3). Two static variables appear to have some strong influence as well, particularly shrubland percentage, which is ranked third in Case 1 and second in Case 2. Precipitation climatology, which represents the spatial distribution of rainfall, is the other static variable with

substantial influence, appearing fourth in Case 1 and accounting for an $R^2$ drop of 0.02 if removed from the model. Besides this variable, no other meteorologically-derived variable, including all the dynamic measures, ranks highly in the presence of Hydroblocks variables. The highest of these is December minimum air temperature, which ranks $19^{th}$ in Case 1. Removing Hydroblocks variables (Case 3) increases the influence of these variables, but they still remain less predictive than the static precipitation climatology and shrubland percentage (ranked second and third), while the dynamic vegetation measures, date

of maximum NDVI and the corresponding value are respectively the first and fourth most influential. NDVI-derived variables otherwise rank $10^{th}$ and $8^{th}$ in Cases 1 and 2.

## 3.3    Field-scale maize yields for Zambia

At the field-scale, RF-predicted maize yields averaged 1557 $\mathrm{kg\,ha^{-1}}$ ($\pm$ 219 $\mathrm{kg\,ha^{-1}}$) for the 2000–2018 period. As expected, we observe higher average maize yields in the northern parts of the country and average lower yields in the southern regions

(Figure 4a), which reflects the spatial distribution of mean rainfall. In terms of the temporal coefficient of variation (Figure 4d), yield variability is highest in the central, southern, and eastern parts of the country, and lowest in northern and northwestern Zambia. In general, there is an inverse relationship between mean annual yields and their coefficient of variation (Figure S2),





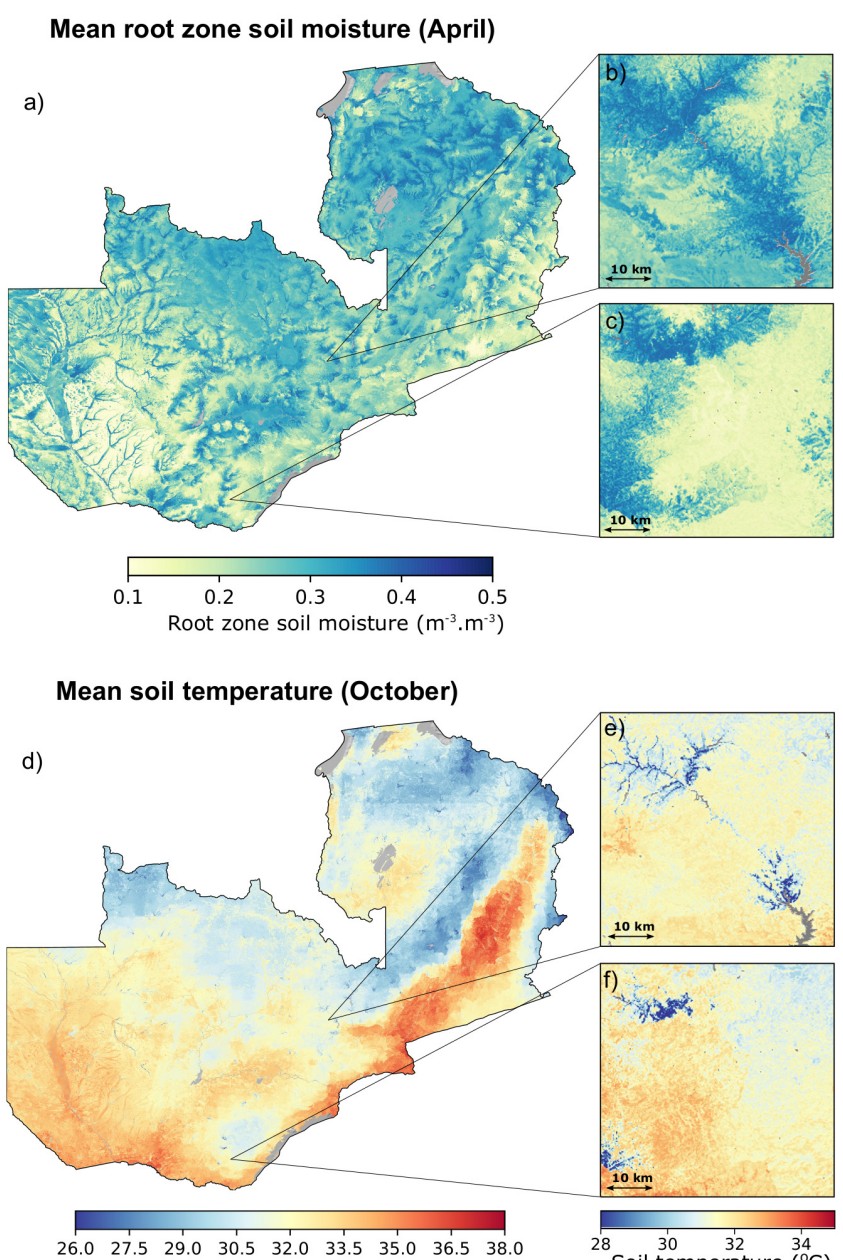

**Figure 1.** April root zone soil moisture and October soil temperature were the most predictive variables in the RF model. Figures a) and b) shows respective mean values (2000–2018), at the 30-m spatial resolution, as simulated by HydroBlocks land surface model. Large lakes are excluded from the simulations (grey areas).




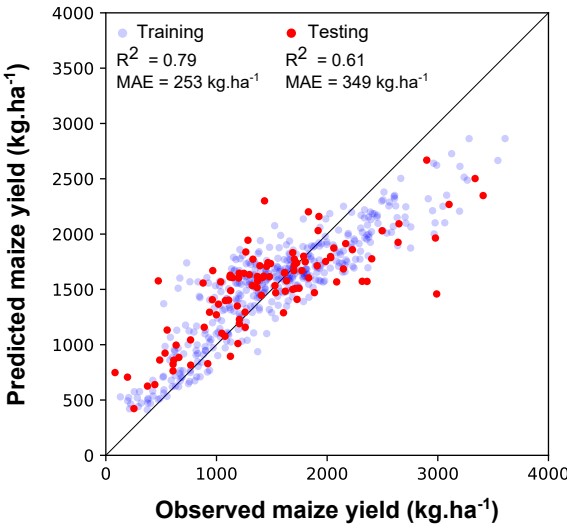

**Figure 2.** Comparison between observed and predicted maize yields at the district-scale. The predictions were obtained by training a random forest model with survey-based maize yield data and fine-scale geospatial environmental datasets.

however, there are several notable departures. For example, a landscape-level view in central Zambia reveals areas of moderate to high yields (Figure 4b) with corresponding moderate to high levels of variability (Figure 4f) due to proximity to rivers and

extensive floodplains, where inundation is frequent. The opposite pattern is seen in Southern Zambia, where portions of the landscape show low yields and relatively low variability (Figures 4c and 4e) due to more consistent dry conditions.

Analyzing the change in maize yields trends for the period 2000–2018 (Figure 4g, 4h, 4i), we found that on the whole maize productivity increased by $3.5 \, \mathrm{kg \, ha^{-1} \, y^{-1}}$ ($\pm \, 4.6 \, \mathrm{kg \, kg \, ha^{-1} \, y^{-1}}$). The gain was more prominent in the northern and central parts of the country, particularly in floodplains (Figure 4i), rising to $15 \, \mathrm{kg \, kg \, ha^{-1} \, y^{-1}}$, while there was little change in southern

Zambia. The productivity also tends to be higher at the locations near floodplains (Figure 4i). Nonetheless, the overall yield gains rates observed were far behind than the rates required to match the $12{,}000 \, \mathrm{kg \, kg \, ha^{-1}}$ Zambia yield potential (Mueller et al., 2012).

During the 2015/2016 El Niño drought, Zambia agricultural production was severely impacted with overall losses across the country. Our field-scale RF-predicted yields were able to capture these losses (Figure 5). The country-wide predicted mean yield

for 2015/2016 was $1514 \, \mathrm{kg \, ha^{-1}}$ ($\pm \, 233 \, \mathrm{kg \, ha^{-1}}$), which represents an overall loss of $84 \, \mathrm{kg \, ha^{-1}}$ ($\pm \, 60 \, \mathrm{kg \, ha^{-1}}$), or 5.3%, relative to 2010–2014 mean. Predicted yield reductions were more severe in southern and southeastern Zambia with losses of $200 \, \mathrm{kg \, ha^{-1}}$ (15%) across much of this area, and reaching as high as $522 \, \mathrm{kg \, ha^{-1}}$ (28%) (Figure 5). These estimates align with those of the 20% losses estimated by the Food and Agriculture Organization (FAO) for this same area during the 2015/2016 El Niño drought (Alfani et al., 2019). However, when evaluating drought impact at the local scales (Figure 5b, 5c), we observe that



**Predictor Importance**

**Figure 3.** The most important predictors for maize yield at the district-scale. The predictors were selected and ranked via Recursive Feature Elimination, with the importance rank shown in terms of delta $R^2$. Results are shown for case 1 (considering all the variables), case 2 (without precipitation and air temperature predictors), and case 3 (without soil moisture and soil temperature predictors). Each color represents different categories of predictors data.

drought impacts yield differently across the landscape. Areas nearby to rivers and floodplains were less prone to crop losses (Figure 5b), given its wetter and cooler soil moisture conditions.

We compared the field-scale yield data aggregated to the district level with the PHS survey district-level yield data. We obtained a Pearson correlation of 0.67 ($R^2$=0.46), and an MAE of 450 $\mathrm{kg\,ha^{-1}}$. In terms of spatial patterns, aggregated field-scale yield and PHS have a spatial correlation of 0.84. This strong spatial relationship is also evident in the PHS and estimated z-scores shown in Figure S3 in the SI. The weak strength and limitations of this aggregated data comparison are discussed in section 4.4





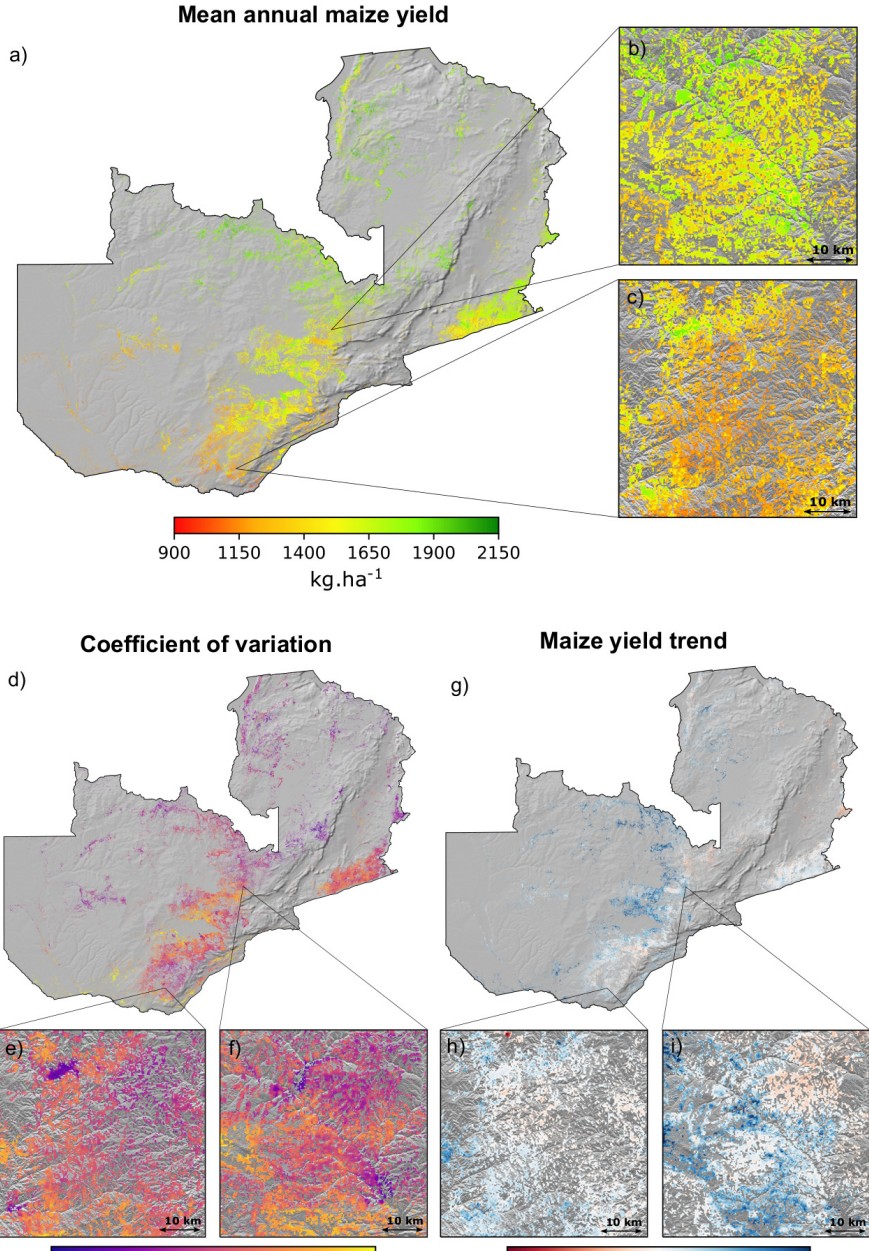

**Figure 4.** Annual maize yields (a), coefficient of variation (d), and maize yield trends (g) for the period between 2000–2018 estimated using a random forest model. Each zoom panel (b, c, e, f, h, i) shows the respective estimates at a 250-m resolution for a 50-km x 50-km area.



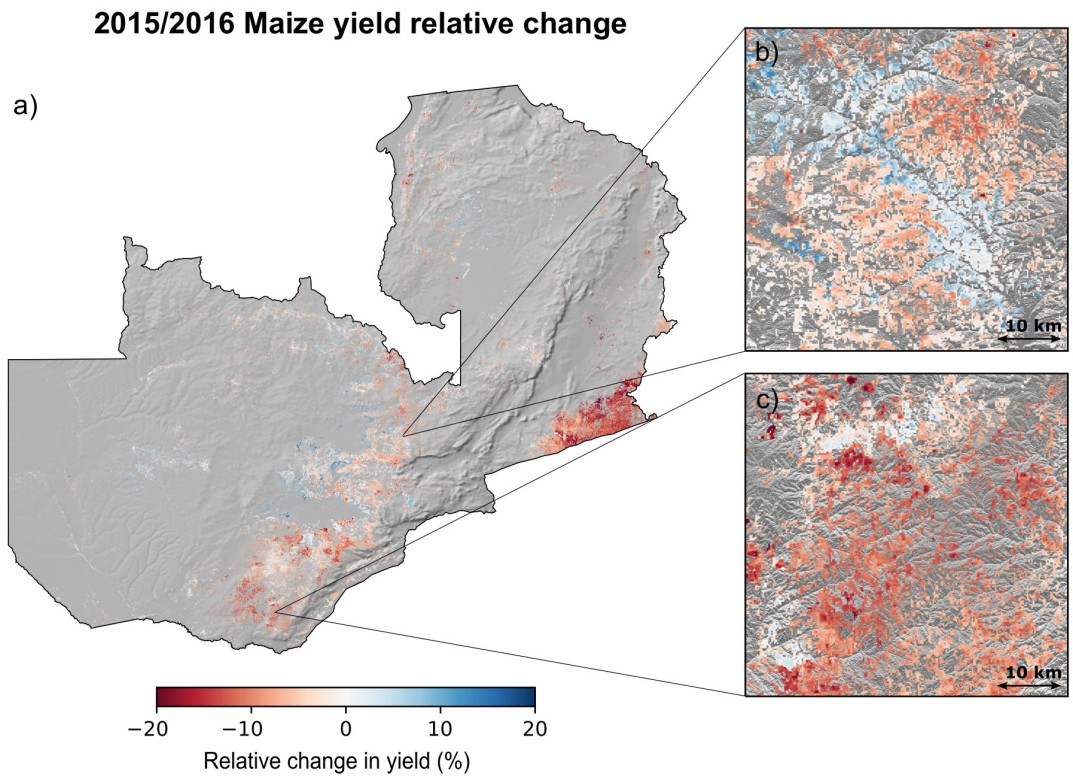

**Figure 5.** (a) Relative change in maize yields (%) for the 2015/2016 season with respect to 2010–2014 mean. Each zoom panel (b, c) shows the respective data at a 250-m resolution for a 50-km by 50-km area.

*Field-scale spatial variability of yields and dominant predictors*

As observed in Figure 5b, droughts impact yields differently across the landscape. To quantify to what extent the spatial variability in the hydroclimate and landscape conditions are driving the spatial variability in the yields, we calculated the spatial correlation between the field-scale yields and the dominant predictors (Figure 6). The spatial correlation was calculated each

year, to assess whether these associations changed inter-annually, and over different locations (the entire country, and for 50 x 50-km boxes in southern, central, and northern Zambia) to assess how the relationships change regionally.

Soil moisture exhibited the largest impact on the spatial variability in yields, with the highest spatial correlation across all three sub-regions, and the entire country (Figure 6a–d). Soil temperature and shrubland percent are negatively correlated with yield. The relative ranking and temporal dynamics of the spatial correlations are generally consistent across regions, although

the strength of correlation varies between regions, with close to zero correlation for most predictors in the central and northern regions. Given its coarser spatial-scale and smoother spatial variability, precipitation climatology showed no spatial correlation with the field-scale yield for each region, but strong correlations at the national scale due to the substantial spatial gradient.

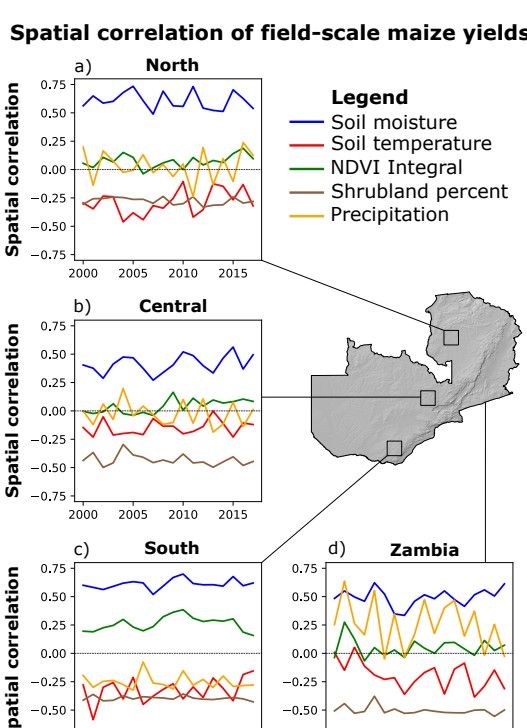

**Figure 6.** Time series of Pearson spatial correlation of maize yields and the most important predictors. Top panels show the spatial correlation within 50 km by 50 km boxes in a) south, b) central, and c) north of Zambia. Panel d) shows the spatial correlation for the entire country.

However, the high inter-annual variability indicates the influence of other factors. With the exception of southern Zambia, NDVI showed a low spatial correlation with yield over time.

### 3.4 The impact of drought on field-scale maize yields

To examine the effectiveness of the 6 potential drought indicators (Table 3), we evaluated the degree of influence that each indicator had on the anomalies of the predicted field scale yields (Figure 7). Overall the relationship between these indicators and yield anomalies was highly non-linear. Soil moisture and precipitation percentiles capture the largest yield losses of all six indicators (Figure 7a, 7d), with both showing substantial negative anomalies below their $25^{th}$ percentile values. Yield losses associated with low soil moisture conditions are larger and more certain than those related to low precipitation. For instance, yield losses associated with the $10^{th}$ soil moisture percentile ($202 \pm 134\ \mathrm{kg\,ha^{-1}}$) were 89% greater than those related to the $10^{th}$ precipitation percentile ($107 \pm 135\ \mathrm{kg\,ha^{-1}}$); orange dashed lines in Figures 7a, 7d. At the $5^{th}$ percentile (red dashed line), average soil moisture-related yield loss ($235 \pm 128\ \mathrm{kg\,ha^{-1}}$) was 26% greater than the yield loss associated with the precipitation ($187 \pm 161\ \mathrm{kg\,ha^{-1}}$), and furthermore has a 20% narrower confidence interval.





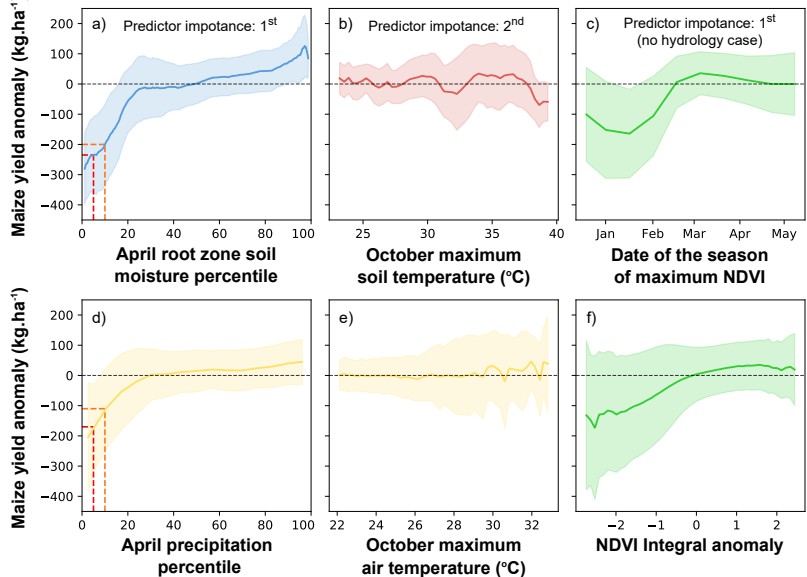

**Figure 7.** The relationship between the field-scale maize yield anomalies with respect to the six drought indicators identified in Table 3. The top row shows the most predictive variables, the bottom row shows the (respective) commonly used variables. Dark lines show the mean values, shades show the standard deviation. Red and orange dashed lines illustrate potential drought thresholds.

Negative NDVI integral anomaly and early NDVI peak date were also associated with yield reductions (Figure 7c, 7f). The strongest and most certain of these was the date of peak NDVI, which resulted in yield anomalies when maximum NDVI occurred before March, with the largest reductions ($164 \pm 146 \, \mathrm{kg\,ha^{-1}}$) occurring for peak dates between January and February. Negative NDVI integral anomalies also showed substantial but more uncertain yield losses. For example, an NDVI integral anomaly of -2 identified a yield loss of $140 \pm 210 \, \mathrm{kg\,ha^{-1}}$.

In contrast to the previous two indicators, there was little variation in yield anomalies associated with soil and air temperature, although the uncertainty in anomalies increased with both measures. However, we observe nearly consistent yield losses with October maximum soil temperatures above 37.5 °C, which is near known critical temperature thresholds for maize (Luo, 2011; Schauberger et al., 2017).

     To gain further insight into the conditions (e.g., hot and dry vs. hot and wet) associated with yield losses, we examined how

the yield anomalies co-varied within pairwise comparisons of drought indicators (Figure 8). As expected from previous findings (Figure 7), soil moisture (Figure 8a, 8c, 8e) and precipitation (Figure 8b, 8d, 8f) percentiles are the dominant influences on maize yield responses. Both indicators show similar patterns, but the yield responses associated with precipitation are noisier. Extreme soil temperature (Figure 8a and 8b) and air temperature (Figure 8c and 8d) only lead to yield losses (red) when the soil moisture and precipitation percentiles are low ($< 25^{th}$). High temperatures under wet conditions (top right corners, Figures

8a-d) show increased productivity (blue). Yield losses occur for the full range of NDVI peak dates when soil moisture and

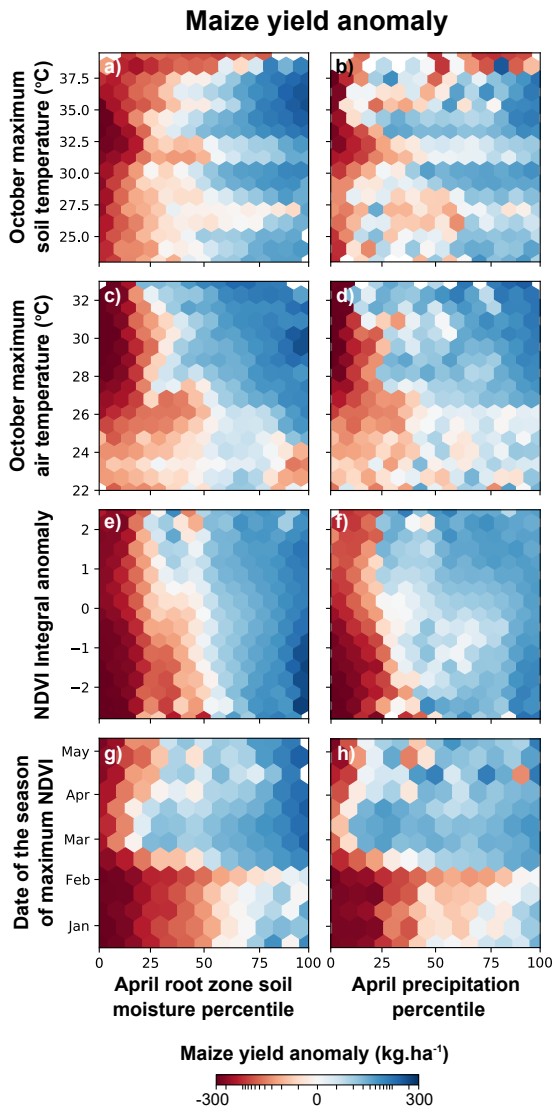

**Figure 8.** Field-scale estimated maize anomalies, as a function of soil moisture and precipitation versus soil temperature, air temperature, date of maximum NDVI, and NDVI integral anomaly. Each hexagon shows the mean yield anomaly, where red (blue) shows overall yield losses (gains). Figure S4 in the Appendix shows the data density of each hexagon.

precipitation percentiles are below $10\text{–}15^{th}$, but when peak date is earlier than March yield losses occur when soil moisture and precipitation fall below their median values (Figure 8g, 8h). NDVI integral anomalies below zero and below-median soil moisture values show a similar relationship with yield (Figure 8e), but this tendency was much weaker when assessed with precipitation (Figure 8f).





## 4 Discussion

### 4.1 Key findings

We presented a modeling framework that combines physically-based hyper-resolution land surface modeling and machine learning for maize yield prediction at the field-scale. Our work advances on previous approaches by integrating field-scale hydrological variables into yield prediction, and by more effectively quantifying yield sensitivity to drought. Our key findings are:

- **Model skill**: The modeling approach that we developed was able to estimate maize yields at district scale with comparable or higher skill ($R^2$=0.61, MAE=349 $\mathrm{kg\,ha^{-1}}$) compared to state-of-the-art approaches based on mechanistic yield models (e.g., Jin et al., 2017; Azzari et al., 2017) and higher skill than standard empirical approaches based on weather variables or vegetation indices (Estes et al., 2013; Zhao et al., 2018).

- **Yield estimates**: The field-scale results showed a mean maize yield of 1557 $\mathrm{kg\,ha^{-1}}$ ($\pm$ 219 $\mathrm{kg\,ha^{-1}}$) across Zambia, with an overall increasing production trend of 3.5 $\mathrm{kg\,ha^{-1}\,y^{-1}}$ ($\pm$ 4.6 $\mathrm{kg\,ha^{-1}\,y^{-1}}$) between 2000–2018. The field-scale yields captured maize losses during the 2015/2016 El Niño drought at similar levels to losses reported by the FAO based on actual yield data (Alfani et al., 2019).

- **Drivers of yield**: We identified soil moisture as the main driver of maize yield variability at both the district-scale and field-scale. At the district-scale, soil moisture was followed in importance by soil temperature, shrubland percent coverage, and precipitation climatology. Time-varying meteorological predictors (precipitation and air temperature) played a minor role. NDVI-based predictors only showed meaningful contribution when soil moisture and soil temperature predictors were absent.

- **Drought impacts**: There is a highly non-linear behavior between drought indices and yield losses. However, consistent maize losses are observed when soil moisture or precipitation drop below the $25^{th}$ percentile. At extreme dry conditions ($5^{th}$ percentile), soil moisture identifies 26% more losses with 21% less uncertainty than precipitation, providing an effective measure of drought impact. Significant yield losses are also predicted when soil temperature exceeded 37.5 °C in the early growing season. Drought impacted yields differently across the landscape (Figure 5), with most of the spatial variability coming from soil moisture (Figure 6).

### 4.2 Drivers of maize yields predictability

*Soil moisture and precipitation*

Soil moisture and soil temperature showed a strong contribution to yield prediction because they represent the water and temperature balances at the root zone, accounting for the soil-water-plant interactions via infiltration, surface and subsurface flow, vertical drainage, water uptake by plants, and evaporation. Consequently, in our sensitivity experiments, in case 1 and



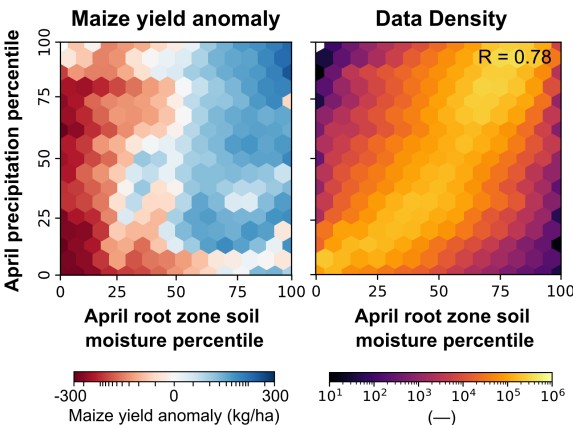

**Figure 9.** Mean maize anomalies and data density as a function of root zone soil moisture and precipitation percentiles.

case 2 the meteorological predictors added only minor improvements to yield prediction beyond the contribution of root zone predictors. Mainly because precipitation does not necessarily translate into water availability for plants. For example, rainfall from intense storms may contribute mostly to runoff, rather than infiltration, leading to a spatial mismatch between where the water is supplied (precipitation) and where water is finally available to plants (soil moisture). Furthermore, whilst temperature drives the atmospheric demand for evapotranspiration, yet it does not reflect the complexity of controls on transpiration, which

are more water-limited (rather than energy limited) in drier regions such as much of Zambia (Berg et al., 2016; Williams et al., 2012; McNaughton and Jarvis, 1991).

    Distinctions between atmospheric and root-zone processes are particularly important at the field-scale, due to different natural scales of variability that influence crops in different ways. Precipitation and air temperature operate mostly at larger scales with most of the spatiotemporal variability coming from large-scale atmospheric circulation and large-scale topographic features

(Grayson and Blöschl, 2001). Soil moisture, on the other hand, operates at multiple scales, influenced by the meteorological conditions, but with most of the spatial variability coming from the heterogeneity of the landscape (topography, soil properties, land cover) (Crow et al., 2012; Famiglietti et al., 2008; Grayson and Blöschl, 2001). By providing soil moisture and soil temperature estimates at a spatial scale closer to its natural scale of variability, HydroBlocks allows for improved yield predictions. This is indicated by comparison of the joint distribution of soil moisture and precipitation percentiles (Figure 9),

which shows that most of the time their condition will be similar (0.78 correlation), yet when they are different (due to the above reasons), soil moisture better captures losses in yield.

    April soil moisture ranked as the strongest predictor of yield. April covers the grain filling stage of the maize calendar in Zambia (∼35 days prior to maturity; Yonts et al., 2008), when soil moisture is critical for plants because of the large water uptake demands (Yonts et al., 2008; Borras et al., 2003). Despite the high greenness of the season (measured by vegetation

indices, such as NDVI, for example), if plants do receive enough water in the reproductive period the cob will not develop well, compromising productivity. Although several studies identified NDVI as the strongest predictors for maize (Table 1 in Funk and





Budde, 2009; Johnson, 2016; Petersen, 2018; Karthikeyan et al., 2020), our sensitive analysis results (Figure 3, case 3) showed that only in the absence of soil moisture and soil temperature variables do NDVI-based variables emerge as strong predictors of yield. This is evident in Figure 8e, 8g which shows that NDVI-based anomalies have little sensitivity to change in yields when

compared to soil moisture. NDVI limitations are discussed in section 4.4.

*Soil temperature and extreme heat*

Soil temperatures in October (sowing period) are also critical in controlling yields. When the rainy season is delayed, extreme temperatures in the early stages can potentially damage seeds prior to their germination (Mulenga et al., 2016), or cause an earlier start of the maize reproductive period, increasing the susceptibility to heat and water stress (Harrison et al., 2011; Hatfield

and Prueger, 2015). However, elevated temperatures in the early season only lead to yield losses when the soil moisture content at the end of the season was low, otherwise, an overall yield gain is observed (Figure 8a). Thus, wet soil moisture conditions (e.g., irrigation) could play an important role in mitigating extreme heat effects (Troy et al., 2015; Thomas et al., 2020) and potentially even increase productivity (Steward et al., 2018), such as observed in Figure 5b. Nonetheless, because of maize's overall sensitivity to elevated temperatures (Lobell and Field, 2007; Lobell et al., 2013), soil temperature above 37.5 °C would

lead to yield losses, despite the wet conditions in the late season (Figure 8a and 8b).

*Static landscape predictors*

In terms of the static predictors, shrubland ranked second as it may represent the heterogeneity of agricultural landscapes. While high cropland percent could be associated with more homogenous agricultural fields (often associated with commercial farming), high shrubland percent may indicate a higher presence of fragmented agricultural areas, reflecting poorer agricultural

landscapes (from a physical and management perspective), and consequently lower yields (Maggio et al., 2018). Figure S5 in the SI illustrates this relationship. Landscape characteristics such as slope and properties of the soil (residual soil moisture, soil water storage capacity, and wilting point), were also identified as important predictors. These predictors control soil moisture dynamics and the water holding capacity of the soil, as more water available from plants for longer increases the likelihood of higher yields.

*Climatological-based predictors*

Climatological precipitation over the growing season also ranked as a strong predictor, as it represents the mean spatial distribution of rainfall. Historically, yields in Zambia are higher in locations with more precipitation during the growing season (Zhao et al., 2018). On the other hand, RFE monthly or seasonal (i.e., dynamic) precipitation did not remain as a model predictor. This could be because soil moisture dominated the predictive power at the sub-annual time scale, but also because of the spatial

scale mismatch between the input data (30-m soil moisture, 1-km climatological precipitation, and 9-km dynamic precipitation predictors). Conversely, climatological air temperature over the growing season did not remain after RFE, but monthly air temperatures in the early season showed predictive importance, highlighting its contribution at sub-annual time scales.



### 4.3 Drivers of maize yield spatial variability

Our results show that there is large spatial variability in yields at national and local scales (Figure 4) that is consistently and

mainly driven by the spatial variability in soil moisture (Figure 6). The spatial variability in soil moisture originates from the interactions between the meteorological conditions and the landscape. For instance, topography controls lateral flows at the root zone, driving the wet (Figure 1b) and cooler (Figure 1e) soil conditions at the river valleys. Soil properties also influence the spatial variability in soil moisture by driving soil moisture dynamics (e.g., infiltration rates), but also thermal properties of the soil (e.g., thermal conductivity). Land cover and vegetation types (Figure ) control land-atmosphere interactions (i.e.,

evapotranspiration), water retention in the root zone, and surface runoff processes. This spatial variability in hydroclimate processes, along with the different farmer management practices, leads to different drought impacts on yields across the landscape. This is evident in the 2015/2016 El Niño drought. Figure 5a, 5b shows that, despite the severity of the event, areas frequently wet and cooler at river valleys are less prone to agriculture losses. This spatial variability in the impact of drought on yields has important implications for decision making (at national and local scales) as traditional coarse-scale drought indices

(and aggregated yield surveys) would have only captured averaged impacts, likely missing the extremes.

### 4.4 Challenges and limitations for field-scale maize yield prediction

Extending the RF-model to predict maize yields at the field-scale is done under the assumption that the model predictors and parameters trained at the district-scale reflect the relationships at the field-scale. However, there are uncertainties in the training data, predictors, and model parameters that lead to uncertainties in the district-level and field-scale level maize yield estimates.

There are also uncertainties in how yields vary at small scales and the physical processes and variability in management that drive this.

*Maize yield training data*

The PHS survey-based dataset, used to train the RF model, was computed by aggregating individual farmer self-reported harvest (number of maize bags) and field areas to the district-level. While aggregating the data provides a more reliable estimate of

yield, as it smooths out the noise and removes outliers, it also constrains the ability of the model to predict high and low yields. Similarly, self-reported field-scale data is also very uncertain and sensitive to errors when estimating yields (harvest/area) (Paliwal and Jain, 2020; Gourlay et al., 2019). Ideally, to improve the capability of modeling yield extremes and to ensure the reliability of the field-scale yield estimates, models should be trained on yield data that was properly measured (in terms of weight and area). Currently, such data is hard to obtain for Zambia (and other countries in the region), except for field trials and

focused research projects. However, by assuming that the RF model trained on district-level data can reflect the major drivers of maize yields, we provide an estimate of field-scale yield variability that is, at least, grounded in reliable district-level estimates. Comparing and validating field-scale yield estimates is also challenging, mainly because of the mismatches between survey-observed and model-estimated yields. For instance, we assume that the PHS survey (district-level yield data) is representative





of the overall yield at each district (i.e., there were no survey sampling issues) and that all 250-m grid cells with at least 50% cropland were maize fields. These mismatches may lead to biases in the aggregated-level estimates.

*Modeling framework and predictors*

Our yield predictions showed a tendency to underestimate high yields (Figure 2). This could be due to above reasons, but also due to the limited training data of high yields ($> 2000 \, \mathrm{kg \, ha^{-1}}$), and a consequence of the RF model averaging many different decision trees (e.g., Baccini et al., 2004; Bourgoin et al., 2018). The underestimation of high yields could also be associated with

the lack of information on farmer management, such as the use of drought-resistant seeds, fertilizers, or irrigation systems, that are not accounted for in the model. The human-driven factors can play a major role in modulating yield outcomes, especially when hydroclimate conditions are not favorable. We expected vegetation index based predictors to improve model capabilities of identifying high yields, however, NDVI only ranked 10th on the RFE (0.01 change in $R^2$, Figure 3). This could be attributed to mixed NDVI signals from other crops (as we only map cropland occurrence, not maize occurrence), infrequent retrievals

(biweekly) with high cloud coverage, and the NDVI tendency to saturate at high LAI values. We also expected socioeconomic based predictors, such as GDP and population, to reflect farmers' access to technology, infrastructure, and markets. However, these predictors were removed by the RFE analysis, showing that yields were not influenced by these socioeconomic variables (similar to Jain, 2007), but mostly influenced by early-season temperature and mid-late season rainfall.

In addition, the uncertainties associated with the strongest predictors could be also reflected in uncertainties in the maize

yield prediction. As most of the temporal and spatial variability in yields comes from the variability in soil moisture and soil temperature (Figure 6), considering the uncertainties from these estimates is fundamental. Uncertainties arise mostly from LSM input data such as soil properties, meteorological conditions, and land cover, as well as model parameterizations. While validation allows us to quantify the reliability of the simulation, in-situ observations are nonexistent in most of the developing world. Given that agricultural yield dynamics are also heavily influenced by human intervention, pathways forward should

consider predictors that better account for human management of crop yields. This can be achieved by accounting for water resources management (e.g., irrigation) when modeling soil moisture dynamics, but also by including additional predictors that reflect farmer decision making (e.g., information on planting dates, seed, and varietal choice, use of pesticides, fertilizers, and machinery, etc.).

## 5 Conclusions

Current drought monitoring often relies on hydroclimate data at coarse spatial resolutions or (infrequent) vegetation index retrievals that do not always indicate the conditions farmers face in the field. As a result, few studies were able to link drought indices to agricultural losses in the field (Karthikeyan et al., 2020; Sutanto et al., 2019). Consequently, decision making (by governments and policy-makers, insurance payouts, water resources management, etc.) based on these indices can often be disconnected from the farmer reality.





With the advancement of hyper-resolution modeling (Wood et al., 2011; Bierkens et al., 2014; Chaney et al., 2016; Vergopolan et al., 2020), a new paradigm has been established that allows field-scale agricultural prediction and drought monitoring. In this work, this is achieved by accounting for the water balance in the root zone, as a critical variable for crop productivity, and by representing soil moisture and soil temperature dynamics at a scale that is representative of farm-scale spatial variability. In specific, we used HydroBlocks model to estimate 30-m soil moisture and soil temperature (1981–2018), and combine these and

other predictors with a Random Forests to model and map 250-m maize yields (2000–2018) across Zambia. To our knowledge, this study is the first to estimate historical field-scale soil moisture and temperature in this region. Our work advances on previous approaches by integrating these variables into yield prediction, and by more effectively quantifying yield sensitivity to drought at the smallholder farm-scale.

    By bridging the spatial scale gap between drought monitoring and field-scale agricultural impacts, this work paves the

way towards applying field-scale soil moisture monitoring to inform agricultural decision-making across scales. Although our approach is complex and involves integrating remote sensing data, hyper-resolution land surface modeling, long-term district level yield data, and machine learning, it can provide the basis for a practical approach to field-scale monitoring that is an improvement over current approaches that are less accurate and at coarse resolution. In addition, it complements and can help to minimize the difficult task of collecting field-based yield data, which is one of the primary limitations for remote agricultural

impact assessments. At the same time, accurate field-scale yield data and information on biophysical parameters (e.g., soils) and farmer-level management practices are still needed to improve and further validate the approach.

*Data availability.*  The data that support the findings of this study are available on request from the corresponding author, N.V.

*Author contributions.*  N.V., L.E., and J.S. conceived the research. N.V. designed and performed the analysis, with support from S.X. in modeling and mapping of maize yields. J.S., L.E, and E.F.W. were responsible for funding acquisition. N.V. led the writing of the paper, and

all co-authors provided critical feedback and contributed to the writing.

*Competing interests.*  The authors declare no competing interests.

*Acknowledgements.*  This work was supported by the National Science Foundation (1832393) and by the National Aeronautics and Space Administration (NNX14AH92G and 80NSSC18K0158).



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
