# Peer review of "Field-scale soil moisture bridges the spatial scale gap between drought monitoring and agricultural yields"

_Hydrology and Earth System Sciences, 2020_

## Referee Comment (RC1) · Hannah Kerner (Referee) · 7 Sep 2020

The manuscript presents maize yield prediction results based on a model that combines hydrological, meteorological, and remote sensing features in a random forest regression. The authors performed feature importance and sensitivity analyses to determine which features influenced maize yield predictions the most and which types of features contributed most to yield prediction accuracy. Overall, the paper is well written and presents useful findings for future studies.

My main criticism for this manuscript is that the random 80%/20% train/test split of yield observations and use of the test set in model optimization are likely overestimating the

performance of the test set compared to the performance that could be expected in practice. The 80%/20% random split across all available yield observations across 70 districts and 8 years does not ensure the test set is independent from the training set. For example, observations of the same district in different years probably have high correlation, as are observations in the same year but different districts. The authors are also optimizing the model for their test set by performing RFE using the test set (they should instead use a third validation set or cross-validation with the training set as was done in 2.3.2). The goal (I assume) of this study is to present a method that can be used to predict maize yields in future years – for the test set to be representative of the performance in this setting, there should be no overlap in years present in the training and test sets (e.g., the training set could include observations from 6 of the 8 years and the test set include observations from the other two years).

Additional comments:

- Why did the authors use 250m MODIS instead of 30m Landsat-resolution NDVI? The latter is much closer to the field scales observed in Zambia and would have the same resolution as the HydroBlocks simulations.

- What computational, time, or cost resources would be required to use the Hy-droBlocks model operationally to predict maize yields in all of Zambia in future years? Is this feasible to do operationally? (Also, note that sometimes the authors write "Hy-droblocks" and sometimes "HydroBlocks".)

- The ESA-CCI 2016 land cover map is used as a cropland mask, and the authors assume all cropland is maize. How valid is this assumption (i.e., what percentage of crops grown in Zambia are typically maize?)? What is the accuracy of this land cover map across Zambia? (I have not seen promising results for this map in Africa.) This could affect the authors' interpretation of shrubland percentage as an important indicator of maize yields.

- The colormap in Figure 9 (right) is missing a title.

- The Figure number is not shown on line 434.

- "Unknowing" should be "Not knowing" on line 60.

- "its" should be "their" on line 2.

- Lines 251-252: the second set of MAE and R-squared values should say they are for the training set (only the test set is mentioned).

---

## Referee Comment (RC2) · Anonymous Referee #2 · 14 Sep 2020

In this work, authors developed a yield prediction system by using a hyper-resolution land surface model HydroBlocks and Random Forest regression. Consequently, authors assessed the sensitivity of predictors towards monitoring droughts. The manuscript is well written with results that can contribute to the ongoing efforts in this area of research. In summary, the manuscript can be accepted for publication once authors address my comments below:

1) It would be nice to see some validation of HydroBlocks simulations. I understand getting root zone soil moisture can be difficult. An alternate comparison with SMAP Level 4 root zone soil moisture product can provide an overview of the quality of HydroBlocks simulations. It may also address, to an extent, one of the limitations of input data uncertainty on yield prediction.

2) How is the calibration of HydroBlocks carried out? Authors may have to provide this information.

3) Apart from underestimation for high yields, there is also overestimation of yields below ~500 kg/ha. Does this indicate that RF produced yield simulations with lower variability than the observed data? Can authors comment on the overestimation?

4) It is surprising to see NDVI not contributing strongly to the prediction of yields. A lot of research on yield prediction depend on NDVI data as a predictor. Can authors throw some more light on this outcome? Is it because there is redundancy in the variance explained by soil moisture and soil temperature compared to that of NDVI towards estimating yields? Since analysis is carried out at monthly resolution, presence of clouds may not be a concern.

5) Spatial scale plays a significant role in driving the soil moisture processes at 30 metre resolution. Can authors assess on the impacts of spatial scales on the yield predictions? Besides, authors may have to describe how various datasets are processed to a consistent spatial resolution.

6) Authors may present time series plots of observed and simulated productivity in addition to the existing results.

7) Minor: Figure 8: How did authors do a pairwise comparison of October maximum temperature and April root zone soil moisture? It is not clear as of how Figure 8 is generated from the description.

8) Minor: Check the units of productivity in the manuscript. At some places, it is given as kg kg/ha instead of kg/ha.

[Figure]

364, 2020.

---

## Author Comment (AC1) · 22 Oct 2020

**Reviewer #1**

The manuscript presents maize yield prediction results based on a model that combines hydrological, meteorological, and remote sensing features in a random forest regression. The authors performed feature importance and sensitivity analyses to determine which features influenced maize yield predictions the most and which types of features contributed most to yield prediction accuracy. Overall, the paper is well written and presents useful findings for future studies.

We thank Dr. Kerner for her thorough assessment and helpful comments.

My main criticism for this manuscript is that the random 80%/20% train/test split of yield observations and use of the test set in model optimization are likely overestimating the performance of the test set compared to the performance that could be expected in practice. The 80%/20% random split across all available yield observations across 70 districts and 8 years does not ensure the test set is independent from the training set. For example, observations of the same district in different years probably have high correlation, as are observations in the same year but different districts. The authors are also optimizing the model for their test set by performing RFE using the test set (they should instead use a third validation set or cross-validation with the training set as was done in 2.3.2). The goal (I assume) of this study is to present a method that can be used to predict maize yields in future years – for the test set to be representative of the performance in this setting, there should be no overlap in years present in the training and test sets (e.g., the training set could include observations from 6 of the 8 years and the test set include observations from the other two years).

We agree with the reviewer's comment that testing/training sampling split per year can provide more robust evaluation statistics. In the revised manuscript, we will include updated statistics using the proposed year-based testing/training sampling approach. We also agree with the reviewer's comment regarding the RFE, in the revised manuscript we will include cross-validation on the RFE feature selection using the year-based testing/training scheme as proposed by the reviewer.

Additional comments:

- Why did the authors use 250m MODIS instead of 30m Landsat-resolution NDVI? The latter is much closer to the field scales observed in Zambia and would have the same resolution as the HydroBlocks simulations.

We agree that Landsat is a better sensor for this purpose, however, MODIS was used instead of Landsat mainly because of the high cloud coverage and a long revisit time of Landsat in Zambia, especially from January to March. Our estimates assessed that cloud coverage between December to February was ~50% in Landsat and ~30% in MODIS (2014-2017). This difference was particularly relevant when calculating the NDVI integrals over the season, in which missing half of the observations could

significantly change the integral results. We will add this information to the revised manuscript.

- What computational, time, or cost resources would be required to use the HydroBlocks model operationally to predict maize yields in all of Zambia in future years? Is this feasible to do operationally? (Also, note that sometimes the authors write "Hydroblocks" and sometimes "HydroBlocks".)

At this first run, the HydroBlocks simulations (1981-2017) took roughly two weeks wall clock time to complete using 200 cores on the Princeton University supercomputing facility (10 nodes, with 20 cores and 128GB per node, in a 2.6 GHz Haswell processor). However, recent developments in the model now allow for faster computations. Although we have not explored this potential yet, as the purpose of these simulations was to understand the historical relationships between yields and droughts, we estimate that ensemble future prediction for 1-5 years could be possible within 1-3 days using the same resources. Therefore, although very powerful, the large computational and big data storage requirements probably wouldn't allow local managers to apply this modeling framework directly, but it can provide a workflow basis for future research development, as highlighted in the conclusion section.

We have replaced "Hydroblocks" with "HydroBlocks" throughout the manuscript. Thanks for the comment.

- The ESA-CCI 2016 land cover map is used as a cropland mask, and the authors assume all cropland is maize. How valid is this assumption (i.e., what percentage of crops grown in Zambia are typically maize?)? What is the accuracy of this land cover map across Zambia? (I have not seen promising results for this map in Africa.) This could affect the authors' interpretation of shrubland percentage as an important indicator of maize yields.

We relied on the assumption that all cropland is maize because of the lack of ground truth for developing a crop type model that would allow us to distinguish maize from non-maize croplands. This assumption is of course incorrect, but maize is by far the dominant crop by planted area in Zambia. According to Zambia's Post Harvest Survey data from 2014-2015, planted maize averaged 60% of the total planted area across all of Zambia's provinces. It's lowest share was 28.4% of planted area in Luapula Province, but the maize share fell below 50% of the area in only three provinces that together accounted for 28% of Zambia's 2014-2015 planted area. In the remaining 72%, the maize share was 69% of planted area.

The odds are thus fairly high that a randomly selected field in Zambia's cropland will be growing maize. Previous studies that used remote sensing to estimate maize yield in Zambia also relied on this same assumption (Azzari et al, 2017). In terms of the impact that this assumption could have on our results, they would likely reduce the accuracy of our yield predictions relative to the PHS dataset, particularly in those districts where the maize share falls below 50%. Mitigating that loss of accuracy, however, is the fact that

our model's predictions are influenced only to a small degree by crop-specific data--since LAI/NDVI makes relatively minor contributions to our model, LAI/NDVI collected from a non-maize pixel will have introduce only small error into the predicted yield. In this sense, our model is essentially predicting for each cropland pixel what the maize yield would be if maize was growing in that pixel.

In terms of accuracy of the cropland mask derived from ESA CCI, we assessed it using an independent accuracy assessment conducted under a separate project. The accuracy assessment was based on a reference sample collected using visual interpretation by three separate raters of both high resolution and Landsat imagery available in CollectEarth, and resulted in 608 validation points where all three raters agreed (444 non-cropland, 164 cropland). Using this sample, the ESA CCI map over Zambia was shown to have user's accuracies of 59% (cropland) and 89% (non-cropland), producer's accuracies of 71% (cropland) and 82% (non-cropland), and overall accuracy of 79%. Since the largest source of error in this map was commission error (41%) by the cropland class, this means that a substantial number of non-cropland areas was predicted to have a maize yield.

Maize yields were calculated at the 250-m pixels for where percentage exceeded 50%. Although our results indicate that shrubland percentage is an important indicator for maize yields, we would like to highlight that, as shown in Figure 5S, high shrubland percentages are associated with low yields (Figure 5S). Thus, despite the low accuracy in land cover classification at the 20-m resolution, the shrubland percentage at 250-m resolution may be accounting for lower productivity at sites that are cropland/shrubland mosaics--which are more likely to be lower-yielding smallholders' fields, and at the same time actual shrublands/cropland mosaics may be more likely to be misclassified as pure cropland (a commission error), and thus explain the negative influence that shrubland percent has on yield. We will add discussion of these important points to the revised paper.

- The colormap in Figure 9 (right) is missing a title.
Corrected.

- The Figure number is not shown on line 434.
Corrected.

- "Unknowing" should be "Not knowing" on line 60.
Corrected.

- "its" should be "their" on line 2.
Corrected.

- Lines 251-252: the second set of MAE and R-squared values should say they are for the training set (only the test set is mentioned).
Corrected.

---

## Author Comment (AC2) · 22 Oct 2020

**Reviewer #2**

In this work, authors developed a yield prediction system by using a hyper-resolution land surface model HydroBlocks and Random Forest regression. Consequently, authors assessed the sensitivity of predictors towards monitoring droughts. The manuscript is well written with results that can contribute to the ongoing efforts in this area of research. In summary, the manuscript can be accepted for publication once authors address my comments below:

We thank the reviewer for their time and thorough assessment.

1) It would be nice to see some validation of HydroBlocks simulations. I understand getting root zone soil moisture can be difficult. An alternate comparison with SMAP Level 4 root zone soil moisture product can provide an overview of the quality of HydroBlocks simulations. It may also address, to an extent, one of the limitations of input data uncertainty on yield prediction.

We agree with the reviewer that evaluation statistics on the soil moisture itself would be interesting. However, there are no available root zone observations for the study area. Regarding the SMAP L4 suggestion, SMAP L4 root zone is at 1 m depth, while HydroBlocks is 1.5 m, but more importantly, their root zone soil moisture product is the result of a model simulation which is subjected to similar, or even more, data uncertainties as HydroBlocks given its coarser spatial resolution. Also, initial validation results in the Continental United States, with in-situ observations, show that at the surface uncalibrated HydroBlocks can perform better than SMAP L4 (Vergopolan et al., 2020). In our opinion, unfortunately, no suitable validation product exists for this region that would allow us to perform on observation drive independent validation of the simulated root zone soil moisture.

Besides the challenges evaluating the absolute soil moisture values, in our approach, the HydroBlocks soil moisture - as the other predictors - are normalized (considering the period between 2000-2018) prior to training the RF yield model. Thus, comparing the absolute root-zone soil moisture values between (2015-2018) may not be representative of the long-term normalized dynamics that are actually being implemented in the yield prediction.

Vergopolan, N., Chaney, N. W., Beck, H. E., Pan, M., Sheffield, J., Chan, S., & Wood, E. F. (2020). Combining hyper-resolution land surface modeling with SMAP brightness temperatures to obtain 30-m soil moisture estimates. Remote Sensing of Environment, 242, 111740.

2) How is the calibration of HydroBlocks carried out? Authors may have to provide this information.
The model is not calibrated, this will be clarified in the manuscript.

3) Apart from underestimation for high yields, there is also overestimation of yields below ~500 kg/ha. Does this indicate that RF produced yield simulations with lower variability than the observed data? Can authors comment on the overestimation?

Indeed the random forest model resulted in a slight overestimation of low yields, we believe this is a consequence of random forest models being somewhat limited in predicting extreme values, as its final estimates are an average of an ensemble of decision trees. This makes it difficult to correctly estimate the observed yields' outlier values as is the case for many RF applications. The discussion on the random forest limitation will also be expanded to discuss the low yields overestimation.

4) It is surprising to see NDVI not contributing strongly to the prediction of yields. A lot of research on yield prediction depends on NDVI data as a predictor. Can authors throw some more light on this outcome? Is it because there is redundancy in the variance explained by soil moisture and soil temperature compared to that of NDVI towards estimating yields? Since analysis is carried out at monthly resolution, presence of clouds may not be a concern.

In Figure 3 (case 3), our results show that NDVI contributes strongly for yields in the absence of hydrological variables. Certainly soil moisture, surface temperature, and NDVI share some variability. In fact, besides predicting yields, NDVI has been extensively reported in the literature as a predictor for soil moisture as well. What our modeling experiment shows, however, is that soil moisture tended to offer more added value than NDVI. In fact, in the RFE experiment, removing NDVI did not result in loss of model performance, whereas removing hydrological variables did reduce model performance. We agree that cloud coverage was not a major limitation for MODIS, but instead, the limitation of visible sensors in capturing under canopy plant-soil-water dynamics with soil moisture and surface temperature, as discussed in the manuscript. We have added this important interpretation of the revised paper.

5) Spatial scale plays a significant role in driving the soil moisture processes at 30 metre resolution. Can authors assess on the impacts of spatial scales on the yield predictions? Besides, authors may have to describe how various datasets are processed to a consistent spatial resolution.

The yield model is trained based on observations and predictors aggregated at the district scale, and under the assumption that this relationship holds at the fine-scale, the model is applied to predict yields at the fine-scales. I assume the reviewer is wondering about the impact of the hydrological variables' spatial scales on the yield prediction. While coarser-resolution hydrological data would not change the performance of the RF much (as the data is trained aggregated to the district-level), we expect it would result in "smoother" field-scale yield predictions. This is expected because hydrology plays an important role in the spatial patterns of the predicted yields, as soil moisture and soil temperature were the strongest predictors (Figure 3). A discussion on the drivers of the spatial variability in yields is presented in section (4.3). We now suggest further

evaluation of the role of spatial scaling for future research in this area in the revised paper.

To obtain a consistent "spatial resolution" for training and predicting, we used a spatial area-weighted average considering only areas where the 30-m land cover is classified as cropland, as described in 3.2.1 section of the manuscript. Thus, despite the model being trained at the district-level, it used the best estimate at the cropland areas for each district. The same approach was used to predict yields at the fine-scales

6) Authors may present time series plots of observed and simulated productivity in addition to the existing results.

A time-series of the observed and predicted soil moisture was not included in the manuscript mainly because PHS training/testing data is not continuous in space and time. Data is obtained for some districts and some years, and thus, time-series comparisons for a single location results in sparse points in time. Consequently, we opted for a scatter plot instead, which allowed us to evaluate the results despite the discontinuities. Nonetheless, in the updated manuscript, we will include a time series of the predicted for some pixels distributed across Zambia.

7) Minor: Figure 8: How did authors do a pairwise comparison of October maximum temperature and April root zone soil moisture? It is not clear as of how Figure 8 is generated from the description.

In Figure 8a, for example, we compared the yield anomaly (of a given season) with the respective October maximum temperature and April root zone soil moisture (for the same season), the hexbins show the mean anomaly yield values when this surface temperature and soil moisture was observed. We will clarify the figure explanation updating the sentences in lines L. 333 to the following: *"To gain further insight into the conditions (e.g., hot and dry vs. hot and wet) associated with yield losses, in Figure 8 we compared how the yield anomalies co-varied with pairwise drought indicators. For example, Figure 8a shows what is the mean yield anomaly associated with its respective October maximum temperature and April root zone soil moisture."*

8) Minor: Check the units of productivity in the manuscript. At some places, it is given as kg kg/ha instead of kg/ha.

Corrected.

---

## Author Response (AR1)

Dear Dr. Wang,

We would like to thank you for handling the manuscript and the reviewers for their thorough assessment and helpful comments. The point-to-point reply to the reviewers was uploaded for discussion on October 22nd, and based on their comments we made the following changes:

1. In the evaluation of the random forest maize yield model, we replaced the randomly selected 20% (testing) and 80% (training) sample split approach for a year-based cross-validation approach. We now report the random forest model testing statistics per year (in which only the other years were used for training) and the average statistics.
2. This year-based cross-validation approach was extended to the recursive feature elimination method, to avoid relying on a single testing set. We updated the workflow in Table S1 and Figure 3.
3. We updated the scatter plot in Figure 2 to show the district-level comparisons per year (with the other years used for model training, and the year in question used for testing). In addition, we included Figure S6 showing the time series of predicted field-level maize yields at different locations across the country.
4. L 165. Included explanation on why MODIS was used in favor of Landsat.
5. L 428. Included a sentence on the influence of cropland/shrubland commission errors on yields.
6. L 137. Included sentence clarifying the HydroBlocks calibration.
7. L 480. Expanded the discussion about why ensemble models (e.g., random forests) have difficulties in predicting extreme low and high yield values.
8. L 490 Discussed the limitations of NDVI in capturing under-canopy plant-soil-water dynamics as represented by data on soil moisture and surface temperature.
9. L 455. Included a discussion on the implications of fine vs. coarse-scale hydrological data on yield estimates.
10. L 345. Improved Figure 8 discussion.
11. Numerous other minor textual corrections.

We uploaded updated versions of our manuscript and supplemental material, as well as a track-changes version of the manuscript.

Sincerely,

Noemi Vergopolan (on behalf of all co-authors)